

# The 12-LOX/12-HETE/GPR31 metabolic pathway promotes tumor-associated macrophage M2 polarization mediated pancreatic cancer development

Ying Yang[1], Yuanzhen Wang[2], Jia Wu[2], Chunxiu Tan[1] and Liya Huang[2]

[1] First School of Clinical Medicine, Ningxia Medical University, Yinchuan, China
[2] Gastroenterology Department, General Hospital of Ningxia Medical University, Yinchuan, China

## ABSTRACT

**Background**. The 12-lipoxygenase (12-LOX)/12-hydroxyeicosatetraenoic acid (12-HETE) pathway is associated with various tumors. M2 macrophages in the tumor microenvironment promote tumorigenesis and progression. However, the role of the 12-LOX/12-HETE/G protein-coupled receptor 31 (GPR31) metabolic pathway and its relationship with M2 macrophages remains unclear in pancreatic cancer (PC).

**Methods**. The expression levels of 12-LOX, GPR31, and 12-HETE were detected in PC and mouse PC models using western blot and enzyme-linked immunosorbent assays (ELISA). *In vivo* and *in vitro* experiments were conducted using the 12-LOX inhibitor ML355 to investigate the role of the 12-LOX/12-HETE/GPR31 metabolic pathway in M2 macrophage polarization and tumor progression through flow cytometry, reverse transcription polymerase chain reaction (RT-PCR), 5-Ethynyl-20-deoxyuridine (EdU) assays, and Transwell experiments.

**Results**. The 12-LOX/12-HETE/GPR31 metabolic pathway is expressed actively in PC. Inhibition of 12-LOX in a mouse model of pancreatic cancer suppressed the expression of this metabolic pathway, retarded tumor growth, and reduced the polarization of macrophages towards the M2 type. *In vitro*, co-culturing PC cell line PANC-1 with macrophages and selectively inhibiting 12-LOX influenced the proliferation, migration, and invasion of PC cells. Inhibiting 12-LOX did not suppress the function of individual PC cells, but it inhibited the development of PC cells co-cultured with macrophages. Moreover, inhibiting 12-LOX reduced the co-cultured M2 macrophages.

**Conclusion**. This study, through *in vivo* and *in vitro* experiments, reveals that the 12-LOX/12-HETE/GPR31 metabolic pathway affects the growth, migration, and invasion of PC by modulating M2 macrophage polarization patterns.

## INTRODUCTION

Pancreatic cancer (PC) is a highly malignant digestive tract tumor. Statistics from The Lancet show a 5-year survival rate of only 10% (*Vincent et al., 2011*). And 90% of tumors are diagnosed in late stages after spreading beyond the pancreas, resulting in over 50%

Corresponding author
Liya Huang, txmbw@126.com

systemic metastasis (*Siegel et al., 2021*), depriving patients of the opportunity for curative surgery. Currently, traditional chemotherapy constitutes the standard treatment for advanced or metastatic PC, but offers only a few months of overall survival benefit (*Von Hoff et al., 2013*). Therefore, further understanding of the potential mechanisms of PC progression is urgently needed for its treatment.

Cancer represents a complex ecosystem, involving tumor cells and numerous non-tumor cells embedded in an altered extracellular matrix. Interactions of non-tumor cells in the tumor microenvironment (TME) play a crucial role in developing pancreatic tumors (*Herting, Karpovsky & Lesinski, 2021*). Tumor-associated macrophages (TAMs) within non-tumor cells represent an immune cell population with both pro-tumor and anti-tumor functions (*De Visser & Joyce, 2023*). TAMs have been shown to drive PC progression by secreting various cytokines such as tumor necrosis factor a (TNF), interleukin 6 (IL-6). *etc* (*Liou et al., 2013*; *Lesina et al., 2011*). They also promote cancer invasion by stimulating blood vessel formation. In addition, TAMs suppress natural killer and T cell functions by expressing non-classical major histocompatibility complex (MHC) class I molecules, co-inhibitory receptor PD-1 ligands, and cytotoxic T lymphocyte antigen 4 (*Liou et al., 2017*).

12-lipoxygenase (12-LOX) is associated with various tumors including esophageal squamous cell carcinoma, gastric cancer, and colorectal cancer (*Mashima & Okuyama, 2015*; *Yang et al., 2019a*; *Yang et al., 2019b*). Studies have found that tumor cells express higher levels of 12-LOX than normal cells and can synthesize 12-hydroxyeicosatetraenoic acid (12-HETE) (*Pidgeon et al., 2007*; *Guo et al., 2011a*). 12-HETE may function through G protein-coupled receptor 31 (GPR31) (*Honn et al., 2016*). In previous studies, it has been found that 12-LOX promotes cancer invasion and metastasis by regulating cell proliferation, angiogenesis, and other factors in prostate and gastric cancer (*Honn et al., 2016*; *Zhong et al., 2018*). Additionally, it has been discovered that 12-HETE enhances tumor cell survival by regulating caspase activity and increasing integrin expression (*Powell & Rokach, 2015*). Furthermore, 12-HETE induces PKC-dependent cytoskeletal rearrangement and secretion of various cytokines, promoting tumor invasion and metastasis (*Powell & Rokach, 2015*; *Kerjaschki et al., 2011*; *Nie et al., 2006*; *Dilly et al., 2013*). However, the role of the 12-LOX/12-HETE/GPR31 metabolic pathway in the occurrence and development of PC remains unclear.

Therefore, this study aims to explore whether the 12-LOX/12-HETE/GPR31 metabolic pathway regulates TAMs polarization to promote the malignant progression and immune evasion of PC. We found that this metabolic pathway is dysregulated in PC and inhibiting 12-LOX can suppress PC development, possibly depending on the reduction of M2 macrophages in TAMs, as shown in both *in vitro* and *in vivo* studies.

## MATERIALS & METHODS

### Clinical samples

The human serum and cancer tissue samples used in this study were obtained from patients undergoing pancreatic resection surgery due to pancreatic ductal adenocarcinoma. Blood

samples were collected before the excision of cancer tissue samples. Informed consent was signed by all blood donors or their relatives. All samples were used only for experimental purposes and research procedures were conducted according to the principles outlined in the Helsinki Declaration. All procedures involving human samples were approved by the Ningxia Medical University General Hospital Scientific Research Ethics Committee (Approval No. KYLL-2022-0045).

## Cell lines

PANC-1 and THP-1 cells were purchased from the Chinese Academy of Sciences Cell Bank (Beijing, China). PANC02 cell was obtained from Cellverse (Shanghai, China). PANC-1 and PANC02 were cultured in H-DMEM (Gibco, Waltham, MA, USA), while THP-1 was cultured in RPMI 1640 (Gibco) containing 0.05 mM $\beta$-mercaptoethanol. All media were supplemented with 10% FBS and 1% penicillin/streptomycin. All experiments were conducted within six months of establishing stable cell lines.

## Mouse model

The 7–8-week-old C57BL/6 male mice were obtained from the Experimental Animal Center of Ningxia Medical University. The mice were housed in the animal facility (humidity 40%–60%, 12-hour light/dark cycle, temperature 20–24 °C), with six mice in a cage measuring 465*300*155 mm. All mice had *ad libitum* access to water and food. Lidocaine gel was applied to the skin under the left axilla of the mice, followed by the rapid subcutaneous injection of $5 \times 10^6$ PANC02 cells using an insulin needle. On the 5th-day post cell suspension injection, the mice were randomly divided into two groups: ML355 was administered *via* intraperitoneal injection (3 mg/kg/day (*Sun et al., 2022*)), referred to as the ML355 group; blank solution served as the solvent control, referred to as the vehicle group (a small amount of lidocaine gel was applied to the injection site before each injection). Based on previous studies (*Arifin & Zahiruddin, 2017*), each group consisted of six mice, with a total of 12 mice used in each experiment. The formula for calculating xenograft tumor volume was $V = \pi \times L \times W \times H/6$ (L: length, W: width, H: height) (*Sun et al., 2022*). Tumor size and mouse weight were monitored every 5 days. The experiment was terminated after 35 days: all mice were anesthetized with 2% isoflurane inhalation, blood was collected from the orbital sinus, and then all mice were euthanized by intraperitoneal injection of pentobarbital sodium at 150~200 mg/kg, followed by a collection of subcutaneous tumors and other organs. During the study, if a mouse's tumor diameter exceeded 15 mm, the mouse was euthanized according to the aforementioned method and excluded from the final analysis. All animal experiments were approved by the Ethics Committee of Ningxia Medical University (Approval No: IACUC-NYLAC-2023-007). All procedures were conducted following the Animal Research: Reporting of *In Vivo* Experiments (ARRIVE) guidelines and regulations. All methods were performed according to relevant guidelines and regulations.

## Cell isolation

Subcutaneous tumors from mice were surgically excised, minced, exposed to a working solution of 1 mg/ml collagenase (Sigma-Aldrich, St. Louis, MI, USA), and dissociated using

**Table 1  Primers used for reverse transcription polymerase chain reaction (RT-PCR).**

| Gene | Forward primer sequences (5′–3′) | Reverse primer sequences (5′–3′) |
| --- | --- | --- |
| Mouse-TNF | GACGTGGAACTGGCAGAAGAG | TTGGTGGTTTGTGAGTGTGAG |
| Mouse-IL12A | CTGTGCCTTGGTAGCATCTATG | CACCAGCATGCCCTTGTCTAG |
| Mouse-NOS2 | GTTCTCAGCCCAACAATACAAGA | GTGGACGGGTCGATGTCAC |
| Mouse-TGFB | CTCCCGTGGCTTCTAGTGC | GCCTTAGTTTGGACAGGATCTG |
| Mouse-ARG-1 | ATGCTCACACTGACATCAACACTCC | GTCTCTTCCATCACCTTGCCAATCC |
| Mouse-Fizz-1 | CCAATCCAGCTAACTATCCCTCC | ACCCAGTAGCAGTCATCCCA |
| Human-TGF | CTAATGGTGGAAACCCACAACG | TATCGCCAGGAATTGTTGCTG |
| Human-IL-10 | ACTGCTCTGTTGCCTGGTCCTC | GCCTTGATGTCTGGGTCTTGGTTC |
| Human-Fizz-1 | GTGTCAAAAGCCAAGGCAGACC | CCAGCTGAACATCCCACGAA |
| Human-12-LOX | ACCAGTTCCTCAATGGTGCC | TCCTCGGATCACGTTGGCT |

a gentleMACS™ dissociator (Miltenyi Biotec, Bergisch Gladbach, Germany). Finally, they were filtered through a 70 µm cell strainer to obtain a cell suspension.

## Co-culture

THP-1 monocytes were differentiated into macrophage-like cells (M0) by incubating with 100 ng/ml phorbol 12-myristate 13-acetate (PMA; Solarbio, Beijing, China) for 48 h. Morphological observations were performed under a microscope to confirm differentiation into macrophages.

M0 macrophages were digested with trypsin and co-cultured with PANC-1 cells ($3 \times 10^4$ cells/cm$^2$) at a 1:1 ratio for 48 h. M0 macrophages were seeded in Transwell permeable cell culture inserts (Corning, Corning, NY, USA; growth area 4.67 cm$^2$; pore size 0.4 µm), while PANC-1 cells were seeded in Costar$^®$ multiwell cell culture plates (Corning). Intervention with ML355 ($30 \times 10^{-3}$ µmol/ml) was provided, with a blank solution used as a solvent control.

## Small interfering RNA knockdown

Lipofectamine 2000 (Invitrogen, Waltham, MA, USA) was used to transfect PANC-1 cells with 12-LOX-targeting small interfering RNA (siRNA) or control siRNA (Shanghai Jikai Gene Chemical Technology Co., Shanghai, China) following the manufacturer's protocol. Knockdown efficiency was verified by RT-PCR analysis at 24 h post-transfection to assess 12-LOX expression levels.

## RNA extraction and real-time quantitative PCR analysis

Total mRNA was extracted from tissue samples or cultured cells using TRIzol reagent (Thermo Fisher Scientific, Waltham, MA, USA). Reverse transcription was done using PrimeScript RT Master Mix (Takara Bio, Kusatsu, Japan) according to the manufacturer's instructions. Real-time quanitative PCR (RT-PCR) was performed using TB Green Premix Ex Taq (Takara Bio) and CFX96 Real-Time PCR Detection System. Primers used are listed in Table 1.

## Flow cytometry

To detect macrophage subtypes, the following antibodies were used for cell staining: Anti-Mouse F4/80 (E-AB-F0995J; Elabscience Biotechnology, Houston, TX, USA), Anti-Mouse CD11b (E-AB-F1081C; Elabscience Biotechnology), Anti-Mouse CD86 (E-AB-F0994D; Elabscience Biotechnology), Anti-Mouse CD206 (E-AB-F1135E; Elabscience Biotechnology), and Anti-Human CD206 (E-AB-F1161E; Elabscience Biotechnology). Blank controls were run in parallel. Flow cytometry was conducted using BD FACSCelesta™ Multicolor Flow Cytometer (BD Biosciences, Franklin Lakes, NJ, USA).

## Western blot

Protein was extracted from tissues and cells. The following antibodies were used: anti-GPR31 (1:1000, ab75579; Abcam, Cambridge, UK), anti-ALOX12 (1:1000, ab168384; Abcam), anti-$\beta$-Actin (1:5000, 20536-1-1AP, Proteintech, Rosemont, Illinois, USA), and anti-Tubulin (1:5000, 11224-1-AP; Proteintech). Following 1-hour incubation with an horseradish peroxidase (HRP)-conjugated $\alpha$-rabbit secondary antibody (1:10000, SA00001-2; Proteintech), protein bands were detected using ECL substrate.

## Immunohistochemistry

Tissues were fixed in 4% paraformaldehyde for 24 h, embedded in paraffin, and sectioned. Deparaffinization was carried out in xylene followed by ethanol gradient hydration. Tissue sections were heated in boiling citrate buffer in a pressure cooker for 20 min. Sections were then treated with 3% $H_2O_2$ at room temperature for 10 min, followed by incubation with anti-GPR31 (1:150, ab150633; Abcam) and goat anti-rabbit IgG H&L (HRP) (1:20000, ab205718; Abcam). A DAB staining kit (Thermo Fisher Scientific) with HRP labeling was used according to the manufacturer's instructions. Samples were mounted and observed under a microscope.

## Enzyme-linked immunosorbent assay

According to the manufacturer's instructions, enzyme-linked immunosorbent assay (ELISA) assay kits (FuSheng and JingMei, China) were used to measure the 12-LOX activity and 12-HETE levels in supernatants from tissue samples or cell cultures.

## Transwell migration and invasion assays

Cells were seeded in RPMI-1640 with or without matrix gel. The culture medium supplemented with 10% FBS was placed in the lower chamber of Transwell permeable cell culture chambers (Corning; growth area 0.33 cm$^2$; pore size 8 $\mu$m). After 24 h of incubation, cells were fixed on the underside of the membrane with 4% paraformaldehyde and stained with 0.1% crystal violet. Migration and invasion cell counts were recorded after capturing stained images under a microscope.

## 5-Ethynyl-2′-deoxyuridine assay

PANC-1 cells were seeded in six-well plates and cultured for 24 h. Following the manufacturer's instructions, cells were cultured with EdU-488 Cell Proliferation Assay Kit (BeyoClic, China), fixed, and then photographed under a fluorescence microscope.

## Statistical analysis

Statistical analysis and processing were performed using GraphPad Prism (version 9.5.0; GraphPad Software, Boston, MA, USA) software, with quantitative data presented as mean $\pm$ standard deviation (SD). Statistical comparisons between groups were performed using a two-tailed Student's $t$-test for normally distributed data, whereas the Mann–Whitney $U$ test (nonparametric) was applied to datasets with skewed distributions. $P < 0.05$ was considered statistically significant.

# RESULTS

## The expression of the 12-LOX/12-HETE/GPR31 metabolic pathway is active in PC

We collected serum specimens from patients who underwent pancreaticoduodenectomy at the General Hospital of Ningxia Medical University due to pancreatic ductal adenocarcinoma (PDAC), as well as pancreatic tumor tissues and their normal tissues. Total proteins were extracted from tumor and normal tissues, revealing low levels of 12-LOX expression in tumor tissues compared to normal tissues (Fig. 1A). However, ELISA testing indicated higher enzyme activity of 12-LOX in tumor tissue (Fig. 1B). In both tumor tissues and the serum of tumor patients, the metabolic product 12-HETE of 12-LOX (*Guo et al., 2011b*) is elevated (Fig. 1D). Furthermore, western blot (WB) experiments indicated an upregulation of the membrane receptor GPR31 for 12-HETE in tumor tissue (Fig. 1A). The immunohistochemistry (IHC) results showed that the mean expression of GPR31 in tumor tissue is higher than the adjacent normal tissue, however, this statistical difference was not significant, possibly due to the small sample size (Fig. 1C).

In conclusion, these results suggested that the expression of the 12-LOX/12-HETE/GPR31 metabolic pathway was active in PDAC patients, which might be related to the occurrence and development of tumors.

## Inhibition of 12-LOX enzyme activity suppresses the development of PC

To investigate the impact of the 12-LOX/12-HETE/GPR31 metabolic pathway on the occurrence and development of PC, we intervened in this pathway using the 12-LOX enzyme inhibitor ML355 in a mouse model of pancreatic cancer. On the 5th day after inoculation of PANC02 cell suspension, ML355 or blank solvent was administered. Mice receiving ML355 intervention were designated as the ML355 group, the control group used blank solvent and was labeled as the Vehicle group (Fig. 2A). Results showed that the tumor size and growth rate in the ML355 group were lower compared to the Vehicle group (Figs. 2B and 2C). Histopathological evaluation using hematoxylin and eosin (H&E) staining and pan-cytokeratin (panCK) IHC demonstrated that the mouse xenograft tissue exhibited characteristic features of carcinosarcoma (Fig. 2D). This suggested that the mouse model can somewhat simulate the development of PC.

By embedding and staining the tumor tissues of mice, we also observed a lower level of GPR31 IHC staining in the ML355 group (Fig. 3A). In addition, ELISA analysis and WB showed that the expression and activity of 12-LOX, the expression of 12-HETE and

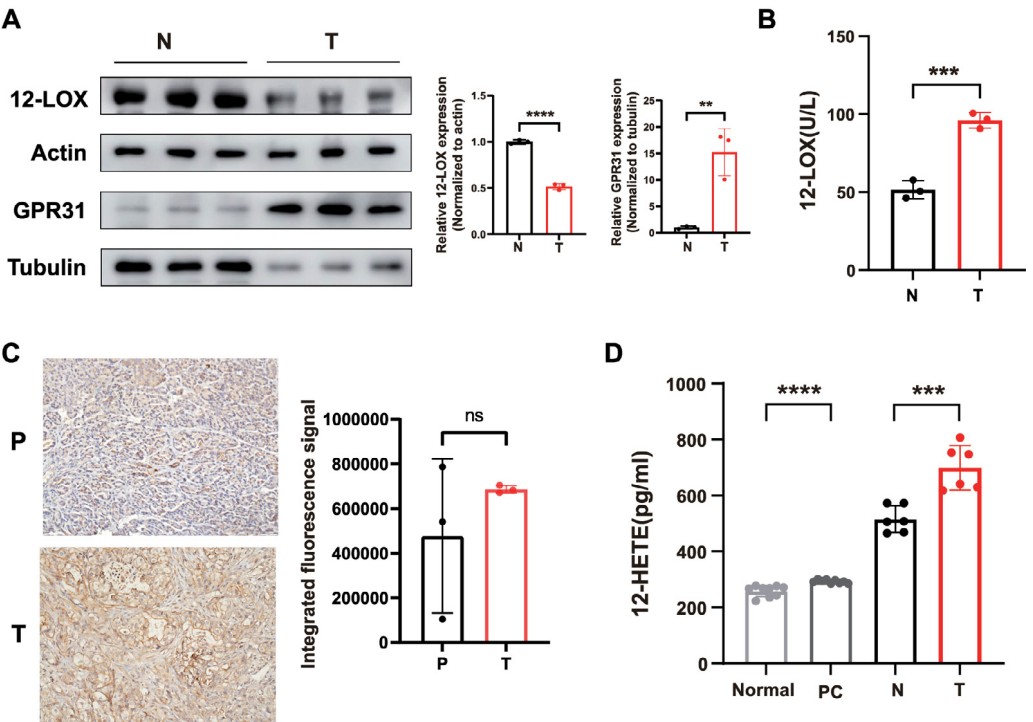

**Figure 1 The 12-LOX/12-HETE/GPR31 metabolic pathway is active in PC.** (A) Western blot (WB) analysis of 12-LOX and GPR31 in PC tumor tissues (T) and normal tissues (N) $n = 3$. (B) Results of ELISA detection of 12-LOX enzyme activity in T and N ($n = 3$). (C) immunohistochemistry (IHC) analysis of GPR31 expression in adjacent normal tissues (P) and T of pancreatic cancer ($n = 3$). (D) ELISA detection results of 12-HETE in the serum of PC patients (PC) and normal human serum (Normal) ($n = 9$), as well as T and N ($n = 6$). Data are shown in mean ± SD; **$P < 0.01$, ***$P < 0.001$, ****$P < 0.0001$, ns, not significant; by Mann–Whitney $U$ test (for the serum groups in D) or by Student's $t$ test for other comparations between two groups.

GPR31 in the tumor tissues of mice in the ML355 group were all suppressed by ML355 (Figs. 3B–3D). Previous studies have demonstrated that the 12-LOX/12-HETE/GPR31 metabolic pathway regulates macrophage infiltration (*Zhang et al., 2018*; *Chung et al., 2019*). Moreover, macrophages have been shown to promote pancreatic cancer progression through the secretion of diverse cytokines (*Liou et al., 2017*; *Von Itzstein et al., 2020*). To examine changes in TAMs within this context, we selected CD86 and CD206 as markers for M1 and M2 macrophages, respectively.

We utilized flow cytometry to detect the subtypes of macrophages in tumor primary cells from different groups of mice. The results indicated that the proportion of CD86[+] cells in the ML355 group increased, while the proportion of CD206[+] cells decreased, with statistical significance (Fig. 3E). In addition, we also measured a series of polarization markers to identify the types of macrophages in tumor tissues. M1 macrophages express interleukin-12A (IL-12a), TNF, and nitric oxide synthase (NOS2), while M2 macrophages express transforming growth factor beta (TGF-$\beta$), ARG-1, and Fizz-1 (*Hansakon et al., 2019*; *Brenot et al., 2018*). RT-PCR analysis showed that inhibiting 12-LOX suppressed the

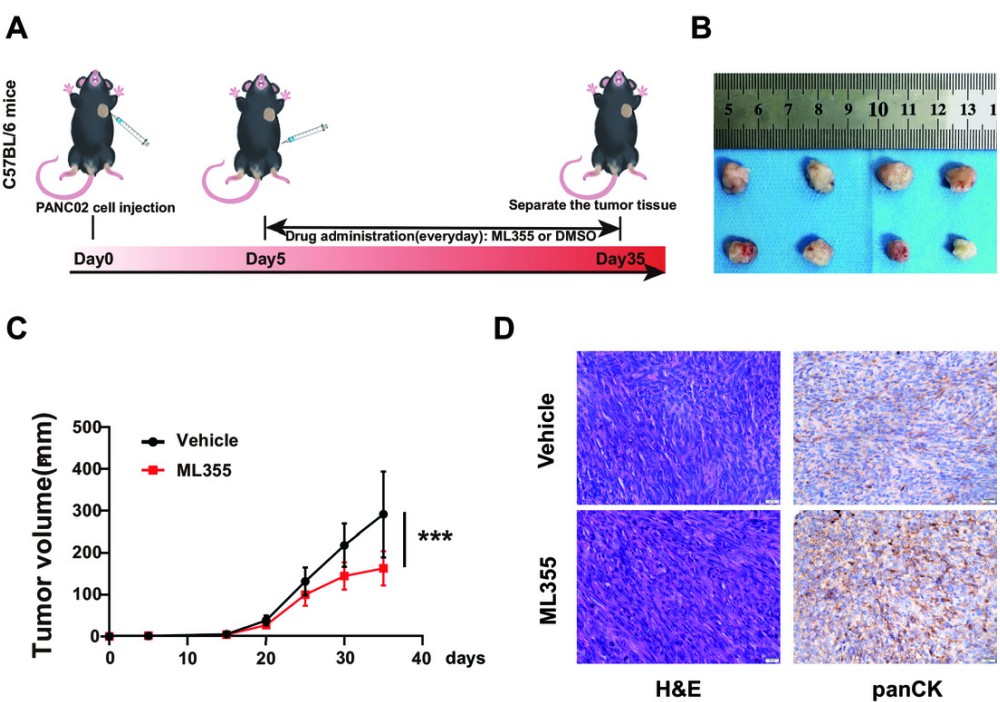

**Figure 2  Inhibition of 12-LOX inhibits the growth of PC xenografts.** (A) Subcutaneous implantation of PANC02 cells in c57 mice, treated with ML355 (ML355 group) or vehicle (Vehicle group) schematic. (B) Representative images of formed tumors. (C) Tumor volumes of mice at different time points post-injection ($n = 12$). (D) Representative images of hematoxylin and eosin (H&E) and panCK staining results in mouse tumors. Data are shown in mean $\pm$ SD; ***$P < 0.001$; by Student's $t$ test for comparations between two groups.

expression of anti-inflammatory factors TGF-$\beta$, ARG-1, and Fizz-1, but the expression of M1 macrophage markers showed both increases and decreases without statistical significance (Fig. 3F).

The above results indicated that intervention of 12-LOX enzyme could intervene in the 12-LOX/12-HETE/GPR31 metabolic pathway, inhibit the growth of xenografts in mice, and also suppress TAMs polarization towards M2 macrophages. This intervention might promote TAMs polarization towards M1 macrophages, but further research is still needed.

### Inhibiting 12-LOX enzyme activity reduces the proliferation, invasion, and migration capabilities of PC cells when co-cultured with macrophages

To investigate whether the involvement of TAMs plays a role in the impact of the 12-LOX/12-HETE/GPR31 metabolic pathway on the development of PC tissue, we co-cultured PC cells with M0 cells and conducted *in vitro* analyses. The cells were divided into four groups: PC group, PC+ML355 group, PC+M0 group, and PC+M0+ML355 group. Results from 5-Ethynyl-20-deoxyuridine (EdU) assays and Transwell assays measuring cell migration and invasion showed no difference in proliferation, migration, and invasion capabilities between the PC group and the PC+ML355 group. The proliferation, migration,

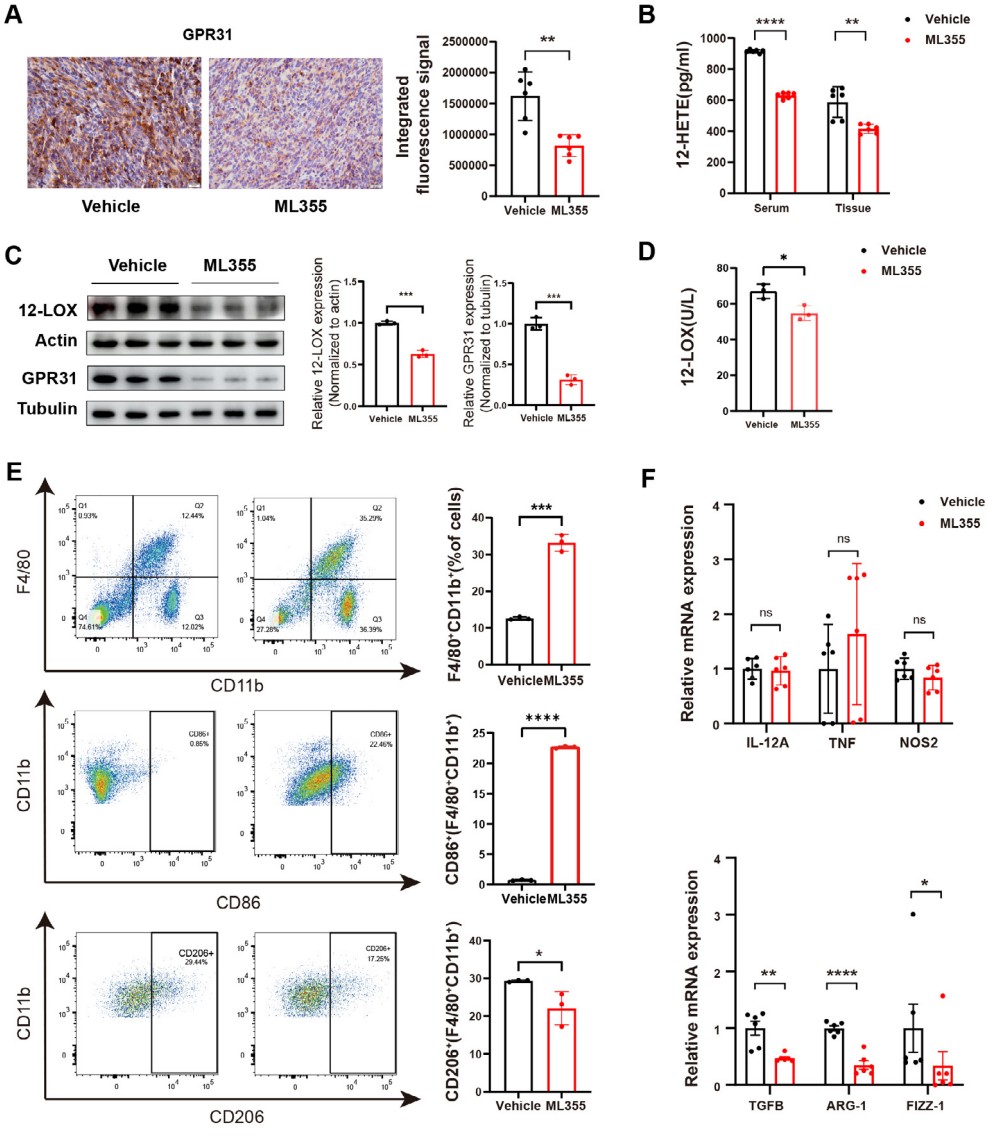

**Figure 3 ML355 inhibits the 12-LOX/12-HETE/ GPR31 signaling axis and reduces the polarization of TAMs to M2 macrophages.** (A) IHC to detect GPR31 expression in ML355 and vehicle groups ($n = 6$). (B) ELISA detection of 12-HETE in mouse serum and tumor tissue of ML355 and vehicle groups ($n = 6$). (C) WB analysis of 12-LOX and GPR31 in mouse tumor tissue of ML355 and vehicle groups ($n = 3$). (D) ELISA assay for 12-LOX enzyme activity in tumor tissue of ML355 and vehicle groups ($n = 3$). (E) Flow cytometry analysis of CD86$^+$ and CD206$^+$ macrophages in ML355 and vehicle groups ($n = 3$). (F) RT-PCR experiment detecting changes in expression of M1 and M2 macrophage markers in ML355 and vehicle groups ($n = 6$). Data are shown in mean $\pm$ SD; $*P < 0.05$, $**P < 0.01$, $***P < 0.001$, $****P < 0.0001$, ns, not significant; by Mann–Whitney $U$ test (for the tissue groups in B and analysis of TNF, TGFB, FIZZ-1 in E) or by Student's $t$ test for other comparations between two groups.

and invasion capabilities of the PC co-cultured with M0 cells were significantly enhanced compared to the PC group. However, compared to the PC+M0 group, the proliferation, migration, and invasion capabilities of the PC+M0+ML355 group were weakened (Figs. 4A and 4B). Flow cytometry analysis revealed a reduction in CD206$^+$ macrophage proportion within the ML355 treatment group (Fig. 4C). Furthermore, RT-PCR demonstrated coordinated downregulation of TGF-$\beta$, interleukin-10 (IL-10), and Fizz-1 (canonical M2 macrophage markers (*He et al., 2022*; *Gong et al., 2022*) in this group (Fig. 4D). To further investigate whether the polarization of M0 macrophages toward the M2 phenotype in the co-culture system is directly influenced by ML355, this study employed siRNA to knock down 12-LOX expression in PANC-1 cells before co-culturing them with M0 macrophages. Flow cytometry analysis revealed no statistically significant difference in CD206$^+$ cell numbers between the groups treated with and without ML355 (Fig. S1C).

Therefore, we speculated that the 12-LOX/12-HETE/GPR31 metabolic pathway may exhibit a macrophage-dependent pro-tumor effect in the TME.

## DISCUSSION

Previous studies have found that 12-LOX/12-HETE is associated with tumor development (*Mashima & Okuyama, 2015*; *Yang et al., 2019a*; *Yang et al., 2019b*; *Pidgeon et al., 2007*; *Guo et al., 2011a*). Our research also found active expression of the 12-LOX/12-HETE/GPR31 metabolic pathway in PC, yet its role in PC occurrence and progression, as well as the underlying mechanisms, remain unclear. TAMs are a crucial component of the TME. M2 macrophages are associated with adverse clinical outcomes in PC, contributing to tumor progression and metastasis (*Hu et al., 2016*). Therefore, we hypothesized that the 12-LOX/12-HETE/GPR31 metabolic pathway affected the polarization pattern of TAMs, promoting the occurrence and development of PC. When using ML355 to inhibit 12-LOX, both *in vitro* and *in vivo* experiments suggested that tumor development is suppressed. Moreover, *in vitro* experiments indicated that this inhibition was partly dependent on the reduction of M2 cells, supporting our hypothesis.

Some research has found that tumor cells have high expression of 12-LOX and can synthesize 12-HETE (*Yang et al., 2019b*; *Pidgeon et al., 2007*). The observed phenomenon in our study—reduced 12-LOX expression yet elevated 12-HETE and GPR31 levels in PC tissues compared to normal counterparts—may result from multiple mechanisms: increased substrate availability (*e.g.*, arachidonic acid), impaired 12-HETE degradation, or enhanced 12-LOX enzymatic activity, among other possibilities. However, integrated with ELISA data confirming upregulated 12-LOX activity in human tumor specimens, we tentatively propose that the activation of this pathway is predominantly driven by hyperactive 12-LOX. Nevertheless, the precise mechanisms underlying this phenomenon, including potential feedback regulation or post-translational modifications of 12-LOX, warrant further investigation. To sum up, we conclude that although 12-LOX expression is low in PC, its enzyme activity is higher, leading to the continued activation of the 12-LOX/12-HETE/GPR31 metabolic pathway in PC. Additionally, a previous study suggested that the 12-LOX/12-HETE/GPR31 metabolic pathway can promote the progression of

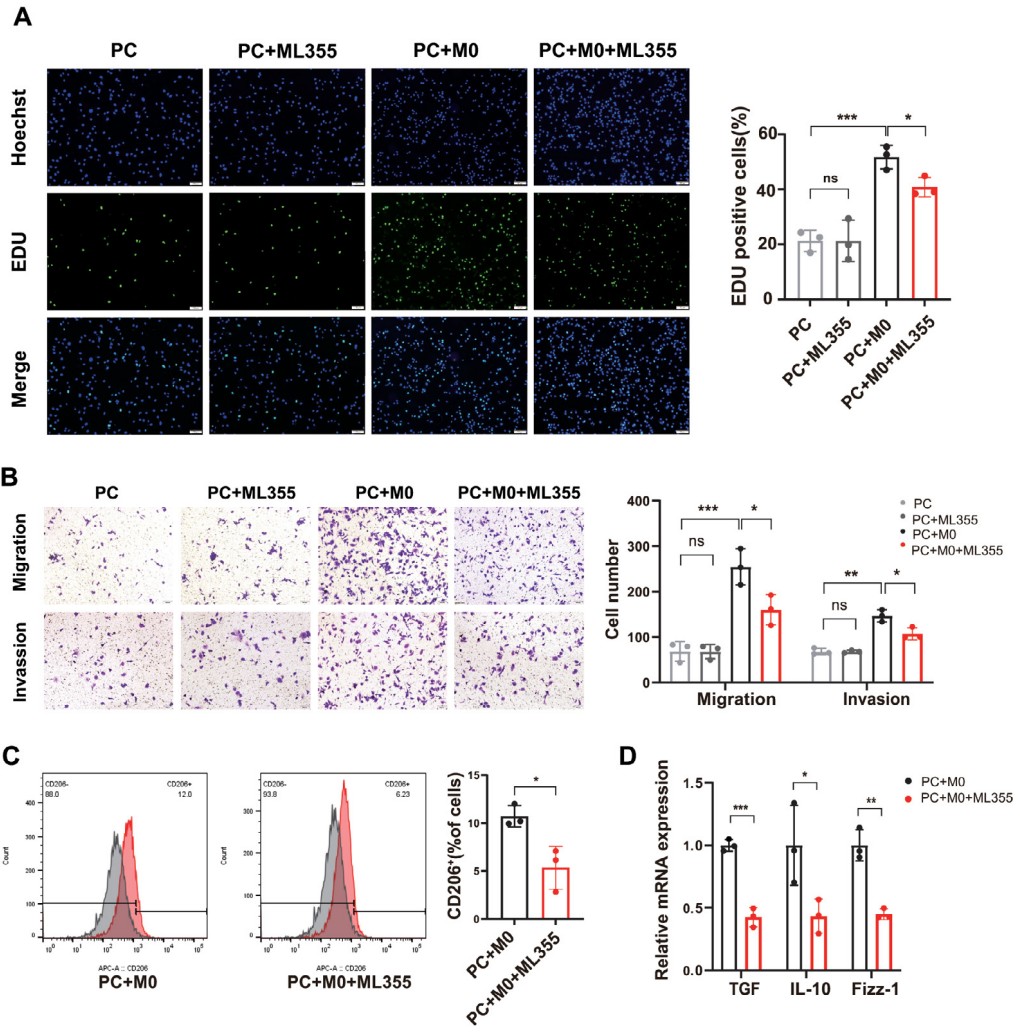

**Figure 4** When PC cells are co-cultured with macrophages, inhibiting the 12-LOX/12-HETE/GPR31 metabolic pathway reduces M2 macrophage polarization, thereby suppressing PC cell proliferation, migration, and invasion. (A) Representative images of EdU detection in PC, PC+ML355, PC+M0, and PC+M0+ML355 groups ($n = 3$). (B) Representative images from Transwell migration and invasion assays in PC, PC+ML355, PC+M0, and PC+M0+ML355 groups ($n = 3$). (C) Flow cytometry analysis of CD206+ macrophages in PC+M0 and PC+M0+ML355 groups ($n = 3$). (D) RT-PCR experiment measuring the expression changes of M2 macrophage markers in PC+M0 and PC+M0+ML355 groups ($n = 3$). Data are shown in mean ± SD; *$P < 0.05$, **$P < 0.01$, ***$P < 0.001$; by Student's $t$ test for comparations between two groups.

prostate cancer (*Kelavkar et al., 2004*). In a mouse model of breast cancer, this signaling axis had been shown to mediate the invasion of intratumorally lymphatic vessels and the spread of lymph node metastasis (*Kerjaschki et al., 2011*). Moreover, it led to more severe hepatocellular carcinoma recurrence in non-alcoholic fatty liver compared to normal liver, and knocking down GPR31 could inhibit its recurrence (*Yang et al., 2019a*). Therefore, in a mouse model of PC, we used the 12-LOX inhibitor ML355. We found that ML355 can inhibit the expression of the 12-LOX/12-HETE/GPR31 metabolic pathway and suppress

tumor development, but cannot reverse tumor initiation. From this, it can be seen that the 12-LOX/12-HETE/GPR31 metabolic pathway may play an important role in the progression of PC.

In further *in vitro* experiments, we found that solely inhibiting 12-LOX in tumor cells did not show inhibitory effects on the migration, invasion, and proliferation of PC cells. Many previous studies have demonstrated that TAMs are recruited to the TME and play various roles in tumor development, including inducing angiogenesis, stimulating tumor cell proliferation and migration, and immunosuppression (*De Visser & Joyce, 2023*; *Fan et al., 2020*; *Laoui et al., 2014*). Our study is in line with previous research findings, where PC cells co-cultured with M0 cells *in vitro* enhanced the proliferation, migration, and invasion capabilities of PANC-1 cells. Therefore, we aimed to further investigate whether the role of TAMs in PC progression is mediated by the 12-LOX/12-HETE/GPR31 metabolic pathway. To address this, we established a co-culture system of PANC-1 cells and M0 macrophages and administered ML355 as an intervention. EdU assay and Transwell experiments revealed that ML355 did not affect the proliferation, migration, or invasion capacities of PANC-1 cells in monoculture. However, when co-cultured with M0 macrophages, ML355-induced inhibition of this metabolic pathway demonstrated significant anti-cancer effects. Collectively, these findings suggest that the anti-cancer activity of 12-LOX/12-HETE/GPR31 pathway inhibition is macrophage-dependent.

Currently, many studies suggest that TAMs polarize into two main phenotypes: M1 and M2 subtypes (*He et al., 2022*). Among them, M2 macrophages, through secretion of various cytokines or exosomes, participate in pancreatic cancer growth, angiogenesis, metastasis, and immune evasion, ultimately promoting the malignant progression of PC (*Von Itzstein et al., 2020*; *He et al., 2022*; *Zhou et al., 2021*). Therefore, we further evaluated the two phenotypes of TAMs. In *in vivo* experiments, when the 12-LOX/12-HETE/GPR31 metabolic pathway was inhibited by ML355, the expression of CD206$^+$ macrophages and mRNA levels of M2 macrophage polarization markers in TAMs decreased; CD86$^+$ macrophages increased, but the mRNA expression of M1 macrophage polarization markers varied. Combining the reduced tumor volume in the ML355 group of mice, we consider the role played by M2 macrophages in this process to be noteworthy. Therefore, in *in vitro* experiments, we examined the situation of M2 polarization-related markers in co-cultured M0 cells and found a decrease in M2 polarization in the ML355 group. Therefore, we consider the anti-cancer effect of inhibiting the 12-LOX/12-HETE/GPR31 metabolic pathway to be macrophage-dependent, with M2 macrophages playing a significant role. Previous studies have demonstrated that 12-LOX/12-HETE promotes interleukin-4 (IL-4) and interleukin-13 (IL-13) secretion to regulate M2 polarization (*Chung et al., 2019*). Furthermore, this metabolic pathway is closely linked to the PI3K/Akt/NF-$\kappa$B signaling pathway in prostate and hepatocellular carcinoma recurrence, and these factors are generally recognized to influence M2 macrophage polarization (*Yuan et al., 2024*; *Wang et al., 2022*). We therefore hypothesize that the observed reduction in M2 macrophage markers may be associated with the activation of these factors or signaling pathways. However, further studies are required to investigate the mechanisms by which this metabolic pathway

influences macrophage infiltration within the tumor microenvironment, which would clarify its role in tumorigenicity and immune response.

This study, for the first time at the protein level, found that compared to normal tissue, 12-LOX is under-expressed but highly active in PC tumor tissue. Furthermore, it was found that the 12-LOX/12-HETE/GPR31 metabolic pathway may not necessarily be directly involved in the development of PC, but rather affects PC development by modulating M2 macrophage polarization. Moreover, the inhibitory effect of ML355 on 12-LOX may be used for the treatment of PC, but this requires further exploration. Additionally, although this study found that ML355 can reduce TAMs polarization towards M2 macrophages, the mechanism by which the 12-LOX/12-HETE/GPR31 metabolic pathway leads to M2 polarization changes is not yet fully understood, and further in-depth exploration is needed.

## CONCLUSIONS

In conclusion, we believed that the 12-LOX/12-HETE/GPR31 metabolic pathway affected the growth, migration, and invasion of PC by regulating the polarization pattern of M2 macrophages. These findings provide new directions for PC therapy.

### Funding

This work was supported by the National Natural Science Foundation of China (No. 82260572). The funders had no role in study design, data collection and analysis, decision to publish, or preparation of the manuscript.

### Grant Disclosures

The following grant information was disclosed by the authors:
National Natural Science Foundation of China: 82260572.

### Competing Interests

The authors declare there are no competing interests.

### Author Contributions

- Ying Yang conceived and designed the experiments, performed the experiments, analyzed the data, prepared figures and/or tables, authored or reviewed drafts of the article, and approved the final draft.
- Yuanzhen Wang performed the experiments, authored or reviewed drafts of the article, and approved the final draft.
- Jia Wu analyzed the data, prepared figures and/or tables, and approved the final draft.
- Chunxiu Tan analyzed the data, prepared figures and/or tables, and approved the final draft.
- Liya Huang conceived and designed the experiments, authored or reviewed drafts of the article, and approved the final draft.
## Human Ethics

The following information was supplied relating to ethical approvals (i.e., approving body and any reference numbers):

All procedures involving human samples were approved by the Ningxia Medical University General Hospital Scientific Research Ethics Committee (Approval No. KYLL-2022-0045).

## Animal Ethics

The following information was supplied relating to ethical approvals (i.e., approving body and any reference numbers):

All animal protocols were approved by the Ningxia Medical University Medical Ethical Committee (Approval No: IACUC-NYLAC-2023-007).

## Data Availability

The raw data is available in the Supplementary Files.

## Supplemental Information

Supplemental information for this article can be found online at http://dx.doi.org/10.7717/peerj.19963#supplemental-information.

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
