# Peer review of "The 12-LOX/12-HETE/GPR31 metabolic pathway promotes tumor-associated macrophage M2 polarization mediated pancreatic cancer development"

_PeerJ, doi:10.7717/peerj.19963_

## Round 0.1 · original submission · Major Revisions

Thank you for submitting your manuscript to PeerJ, which has been through the peer-review process. Reviewer comments are below. When revising your manuscript, please carefully consider all issues mentioned in the reviewers' comments: please outline every change made in response to their comments and provide suitable rebuttals for any comments not addressed. Please note that your revised submission may need to be re-reviewed.

**Language Note:** The review process has identified that the English language must be improved. PeerJ can provide language editing services - please contact us at [email protected] for pricing (be sure to provide your manuscript number and title). Alternatively, you should make your own arrangements to improve the language quality and provide details in your response letter. – PeerJ Staff

·

Basic reporting

In this manuscript, Yang et al reveal that the 12-LOX/12-HETE/GPR31 metabolic pathway induces the growth, migration, and invasion of pancreatic cancer in a M2 macrophage-dependent manner. This study contains interesting findings and is valuable for the understanding of the role of the 2-LOX/12-HETE/GPR31 metabolic pathway in pancreatic cancer. However, a minor revision has to be done before this manuscript can be accepted for publication in PeerJ. The detailed suggestions are listed below.

(1) In lines 191-192, the author didn’t explain why total proteins extracted from tumor and normal tissues revealed low levels of 12-LOX expression in tumor tissues compared to normal tissues, which is quite the opposite from the ELISA test and IHC result.

(2) In lines 220-221, the reason for emphasizing the study on TAMs is not adequate.

(3) In lines 226-227, references are needed to support “M1 macrophages express IL-12a, TNF, and NOS2, while M2 macrophages express TGF-, ARG-1, and Fizz-1”.

(4) Some references are out of date, such as references 4 and 19.

(5) Few grammar mistakes should be corrected, such as “In addition, considering that the development of tumors is accompanied by changes in immune cells in the TME6. To investigate the changes in TAMs in this process, we selectively chose the specific markers CD86 and CD206 for M1 and M2 macrophages, respectively. ” in line 220-221.

(6) The language of this manuscript needs to be polished by a native English speaker or a professional language editing tool.

Experimental design

-

Validity of the findings

-

·

Basic reporting

This study investigates the role of the 12-lipoxygenase (12-LOX)/12-hydroxyeicosatetraenoic acid (12-HETE)/GPR31 metabolic pathway in pancreatic cancer (PC) development, focusing on its interaction with tumor-associated macrophages (TAMs). The research found that this pathway is actively expressed in human PC tissues and patient serum. Using a mouse xenograft model, inhibition of 12-LOX with ML355 suppressed tumor growth, reduced pathway marker expression, and decreased the polarization of TAMs towards the M2 phenotype. Critically, in vitro experiments demonstrated that while inhibiting 12-LOX alone did not affect PC cell proliferation, migration, or invasion, it significantly suppressed these malignant behaviors when PC cells were co-cultured with macrophages, an effect correlated with reduced M2 macrophage polarization in the co-culture system. The study concludes that the 12-LOX/12-HETE/GPR31 pathway promotes PC progression, at least in part, by modulating TAM polarization towards the pro-tumorigenic M2 phenotype, suggesting this pathway could be a potential therapeutic target.

Experimental design

1. A significant discrepancy arises from the data presented, which indicates that while 12-LOX protein expression is reportedly decreased in PC tumor tissues compared to normal tissues (based on Western Blot analysis), its enzymatic activity is paradoxically increased (based on ELISA results). The authors attempt to reconcile this in the discussion by suggesting "low expression but high activity," yet this explanation lacks sufficient mechanistic support or evidence. The activation of the 12-LOX/12-HETE/GPR31 pathway despite reduced protein levels warrants a more profound investigation into potential causes, such as post-translational modifications, specific enzyme activation mechanisms, altered levels of endogenous inhibitors, issues with Western Blot antibody specificity or efficacy, or sample processing variability. Alternatively, this contradiction and its potential impact on the findings should be acknowledged more rigorously in the discussion. The current explanation is overly simplistic and fails to be convincing.

2. ML355 is employed as an inhibitor primarily targeting 12-LOX enzymatic activity. However, Western Blot results presented for the mouse model suggest that ML355 treatment might also reduce 12-LOX protein expression levels. This finding potentially conflicts with the observation of low basal 12-LOX protein expression reported for human PC tissues and the established role of ML355 as an activity inhibitor. Clarity is required regarding whether ML355, under the experimental conditions used, solely inhibits enzyme activity or also impacts protein expression. If the latter is true, the underlying mechanism must be elucidated, as this is crucial for interpreting the inhibitor's effects. Furthermore, the observed reduction in GPR31 protein levels and immunohistochemical staining following ML355 treatment in vivo indicates downstream pathway inhibition, but it needs to be clarified whether this is a direct consequence of ML355 action or an indirect effect secondary to reduced ligand production or other pathway alterations.

Validity of the findings

1. There are potentially confusing statements regarding the co-culture experiments that require clarification. The manuscript suggests that ML355 inhibits PC cell proliferation, migration, and invasion specifically under co-culture conditions (when compared to the co-culture control group). However, the text later states that inhibiting 12-LOX in monocultured tumor cells did not show inhibitory effects (when compared to the monoculture control). While this supports the overall conclusion of macrophage-dependent effects, the presentation could be clearer. To robustly establish this dependency, the authors should ensure the interpretation is unambiguous, potentially strengthening the point with controls addressing whether ML355 affects macrophage viability or function directly in the co-culture system, independent of 12-LOX in PC cells (if macrophages also express the target).

2. While the study effectively demonstrates a correlation between ML355 treatment (and thus, inferred inhibition of the 12-LOX/12-HETE/GPR31 pathway) and reduced M2 macrophage polarization markers based on the in vivo and in vitro findings, a notable mechanistic gap exists. The direct molecular steps linking the inhibition of this specific metabolic pathway to the observed shift away from M2 polarization remain unestablished. Elucidating the intermediate signaling events (e.g., downstream transcription factors, alterations in key cytokine profiles within the co-culture supernatant driving polarization, or specific intracellular signaling pathways modulated by GPR31 activation/inhibition in macrophages) through additional experiments would be necessary to substantiate the causal claim beyond correlation.

Additional comments

1. Figure legends should be comprehensive, standalone descriptions. Please ensure that each figure legend clearly states the number of biological replicates (N value) for each group, defines the error bars (SD or SEM), and specifies the statistical test performed for the data shown in that specific panel.

2. Please provide a brief justification for the specific concentration of ML355 used in vitro and the dose used in vivo, confirming their relevance and efficacy in the context of pancreatic cancer cells/models beyond citing the provided reference.

---

## Round 0.2 · accepted · Accept

The reviewers' concerns have been addressed. My recommendation is to accept.

·

Basic reporting

The author answered reviewers' questions thoroughly, and provided solid and reliable results.

Experimental design

I believe the revised article meets the standard of this journal, and no further improvements are needed.

Validity of the findings

The findings are valid.

Additional comments

No comment.

·

Basic reporting

no comment

Experimental design

no comment

Validity of the findings

no comment

Additional comments

no comment